# Knowledge Distillation for Predicting Varying Environment Maps from Single Images

## Abstract

We present a learning-based method for estimating view-dependent environmental lighting from a single image. Our method learns to distill knowledge from a differentiable geometry and texture decomposition framework. The goal is to directly predict the environment map from an input image using a neural network and thus bypass the need for solving iterative optimization. We propose a new physically-based strategy that decouples the illumination color and distribution of a local light probe given by a sampled pixel on the input image. The experiments show that our proposed method can train a neural network to efficiently derive the environment map of comparable or even higher quality from a single image in under a second, a significant improvement over the time-consuming optimization-based alternatives that require a few minutes to obtain the results.

## 1 Introduction

The task of 3D reconstruction and reflection decomposition from multi-view images is twofold: First, one needs to reconstruct the geometry of the target into common 3D-modeling formats such as point clouds or meshes, and second, the material has to be decomposed into attributes like normal, roughness, metallic, diffuse factor, and specular factor, which are essential for calculating the reflection on the target with the illumination. Typically, this task assumes a set of images captured from different camera poses under fixed lighting conditions. The geometry, material, and illumination are jointly obtained after the decomposition.

An extended task would be assuming a less-constrained and ill-posed setting, for example, decomposition under varying illuminations, which entails that for images in the collection, both the viewing directions and the illuminations are varying. When evaluating such a complex task of simultaneous 3D reconstruction and reflection decomposition from multi-view varying-illumination images, one would need to synthesize novel views and render the target under new illumination conditions to verify the reconstruction quality and decomposition accuracy.

Illumination estimation endeavors to extrapolate the comprehensive environmental lighting from a segmented portion of a scene. An advanced exploration of this topic would be to integrate it with inverse rendering techniques. Through inverse rendering, one can infer the environmental illumination solely from a single-view image of the designated target. Subsequently, the derived environment maps can be re-lit on different objects. The rendering results are compared with the ones lit by ground-truth illuminations to verify the accuracy of estimation.

In this work, we consider solving the integration of the aforementioned tasks: given an image collection of the target captured under varying illuminations and different viewing directions, we start with the reconstruction and decomposition to obtain the geometry and material, and then we train a neural network to learn the relation between the varying environmental illuminations and the lighting exhibited on the target based on the decomposed geometry and material. As a result, we obtain a neural network that can efficiently predict the varying environmental illumination directly from any given image of the target.

To achieve reconstruction and decomposition, we adopt a state-of-the-art optimization-based decomposition tool, *nvdiffrec* (Munkberg et al., 2022), originally designed for scenarios with fixed illumination. We apply it to our varying illumination settings to derive the geometry and material of the target from multi-view varying illumination images and then relax nvdiffrec to produce an

individual environment map for each image instead of a fixed environment map for all images. Regarding the estimation of lighting, we propose a new strategy for predicting the environment map by decoupling the illumination color and distribution of local light probes from sampled pixels. The light probes' decoupled illumination color and distribution can then be re-combined to generate the complete environment map. We train a neural network to learn the illumination color using a tailored loss to ensure improved performance and design the distribution following physically-based rendering to generate realistic results. Given a new input image of the target captured under an unknown illumination, our method can bypass the need for solving iterative optimization by nvdiffrec, and, instead, directly predicts the environment map using the neural network trained with the distilled knowledge from nvdiffrec. Our proposed method can derive the environment map within a second without solving optimization.

## 2 RELATED WORK

**3D Reconstruction.** Recent advancements in view synthesis and 3D reconstruction have been impressive and exciting, particularly with the introduction of the neural radiance field concept by NeRF (Mildenhall et al., 2020). This innovative approach leverages a Multilayer Perceptron (MLP) to decode the 3D radiance space from multi-view 2D images via incorporating positional encoding to capture intricate high-frequency details. Successive methods have refined both the quality of reconstruction and the efficiency in training and inference through diverse representations. For instance, Mip-NeRF 360 (Barron et al., 2022) enhances the encoding technique to deal with the issue of aliasing and hence can produce finer details for view synthesis. Plenoxel (Fridovich-Keil et al., 2022) and DVGO (Sun et al., 2022) introduce faster convergence mechanisms for deriving the radiance field by directly optimizing on a voxel grid with trilinear interpolation and thus significantly reducing computation time. Contemporary research has been broadening the scope by exploring extensions in characterizing not just the color and density but also the geometry, material, and illumination.

**Decomposition Under Fixed Illumination.** An inherent limitation of the radiance field is its inability to accommodate variations in lighting conditions robustly, which causes challenges in scenarios that require relighting under different illumination. The neural radiance field predominantly captures the color radiance within a given space yet neglects the mutual influences between incident radiance and the target surface's reflection. NeRV (Srinivasan et al., 2021) thus introduces a neural reflectance field to distinguish between diffuse and specular reflections. Building upon the foundational idea of the reflectance field, NeRFactor (Zhang et al., 2021b) implements a decomposition approach that learns to estimate the Bidirectional Reflectance Distribution Function (BRDF) reflection using multiple MLPs. Note that most neural field-based methods focus on refining view synthesis quality but often neglect the actual illumination. PhySG (Zhang et al., 2021a) applies the notion of 3D reconstruction via neural fields to inverse rendering and seeks to generate the environment map concurrently during the 3D reconstruction process. While many have employed neural rendering within neural field representation to compute the rendered radiance, PhySG introduces an SG rendering method that approximates the varied incident light by the combination of spherical Gaussians around the environment map. While the representation of lighting in PhySG is flexible enough to calculate the reflection on different textures of the target, the produced environment map is often blurred and is of low resolution due to the limitation of Gaussians. Following the physically-based rendering, a more advanced process of decomposition and inverse rendering called *nvdiffrec* is proposed by (Munkberg et al., 2022). The geometry and material of the target are decomposed into triangular mesh and 2D texture, which are popular formats in most rendering engines. To produce illumination in high resolution, Munkberg et al. (2022) propose a differentiable split-sum approximation of environment maps with an optimization-based representation. Their follow-up method *nvdiffrec-mc* (Hasselgren et al., 2022) replaces the split-sum approximation with an important lighting sampling for Monte Carlo rendering and denoising. The Monte Carlo rendering improves the decomposition quality and handles the material issue compared with the previous version *nvdiffrec*.

**Decomposition Under Varying Illuminations.** While most methods pursue more precise decomposition under fixed illumination, NeRD (Boss et al., 2021a) brings out a more ill-posed but realistic assumption. The images in their collection are captured under varying illuminations, which means for each image, not only does its viewing direction differ, but also the image is captured under varying and unknown illumination. Following the general architecture of NeRF (Mildenhall et al., 2020)

with the decomposition network for basic view-dependent color and BRDF reflection, the learned latent space encodes different lighting for varying illuminations. Neural-PIL (Boss et al., 2021b) proposes a pre-integral lighting network to match the illumination latent code and further enhance the efficiency and accuracy of rendering. The application of their methods includes *i*) relighting the target with different illuminations by editing the latent code or *ii*) lighting estimation from a single image captured under unknown illumination by optimizing a latent code.

**Lighting Estimation.** A more challenging extended task to the topic of decomposition would be estimating the lighting from a single image. The task usually requires the ability to estimate the complete environment map from partial information about the illumination. ALP (Yu et al., 2023) attempts to assume a more difficult but useful condition to acquire the high-resolution environment map in the wild. They consider a scenario that the target is some highly-reflective common object like a metal can, with the reconstruction from the laboratory for more precise geometry mesh and the optimization-based method adopted from nvdiffrec-mc (Hasselgren et al., 2022) for material and illumination. They obtain environment maps that are accurate enough to generate relighting effects on other objects with different textures. Similar tasks around the lighting estimation topic include StyleLight (Wang et al., 2022) and Deep Parametric (Gardner et al., 2019). Another variant on this topic is spatially varying lighting estimation. Rather than estimating the illumination from an object, the targets are usually the entire scene. Without the common assumption of location-agnostic deep lighting environment maps, illuminations are estimated depending on the locations around the scene. Some spatially-varying lighting estimation methods includes (Garon et al., 2019), (Choi et al., 2023), (Li et al., 2020), and (Srinivasan et al., 2020).

## 3 METHOD

To decompose a scene based on lighting reflection, our method leverages a physically-based rendering approach that follows the Bidirectional Scattering Distribution Function (BSDF) reflection. To predict the unknown illumination from the reflection on the target, we adopt the split sum approximation of specular reflection for simulating the illumination colors and distributions of local light probes across various directions.

Due to the highly ill-posed settings for simultaneously *i*) decomposing the scene into geometry and material and *ii*) learning the illuminations from the target, we divide our method into two phases of learning. In the first phase, we focus on decomposing the scene to obtain the triangular mesh of geometry, the 2D texture of material, and the environment maps for the illuminations. In the second phase, we train an illumination MLP to predict the color of light probes directly from the target and produce the complete environment map.

### 3.1 PHASE I: DECOMPOSITION UNDER VARYING ILLUMINATIONS

In the first phase, we decompose the scene into geometry, material, and illuminations. The input data in our assumption are multi-view images captured under varying unknown illuminations. After the decomposition, we obtain the triangular mesh and 2D texture. Also, for each view of the input image, we optimize a corresponding environment map at the same time to learn the geometry and material. The process of the first phase is summarized in Figure 1.

#### 3.1.1 RENDERING EQUATIONS

The decomposition is achieved by nvdiffrec (Munkberg et al., 2022), which is a powerful tool built on a differentiable physically-based renderer. The rendering equations follow BSDF reflection:

$$L(\omega_o) = \int_{\Omega} L_i(\omega_i) \, f(\omega_i, \omega_o) \, (\omega_i \cdot \boldsymbol{n}) \, d\omega_i \,, \tag{1}$$

where $L(\omega_o)$ is the reflected light, $L_i(\omega_i)$ is the incident light, $f(\omega_i, \omega_o)$ is BSDF, $\boldsymbol{n}$ denotes the normal vector, $\omega_i$ denotes the incident direction of light, $\omega_o$ denotes the reflected direction of light, and $\Omega$ is the hemi-sphere centered at $\boldsymbol{n}$. Note that the specular reflection in nvdiffrec is represented by split sum approximation:

$$L(\omega_o) \approx \int_{\Omega} f(\omega_i, \omega_o) \, (\omega_i \cdot \boldsymbol{n}) \, d\omega_i \int_{\Omega} L_i(\omega_i) \, D(\omega_i, \omega_o) \, (\omega_i \cdot \boldsymbol{n}) \, d\omega_i \,, \tag{2}$$

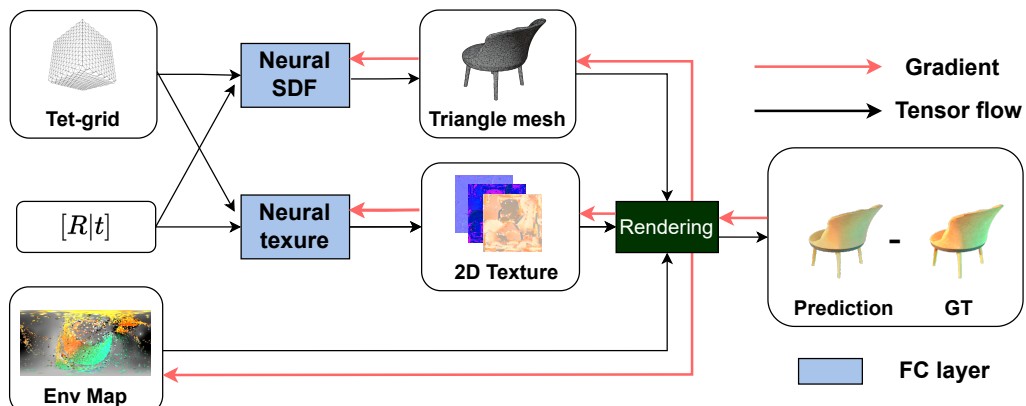

Figure 1: An overview of the first phase in our pipeline.

where $D$ is a function representing the GGX normal distribution (NDF) as mentioned in nvdiffrec. In the second phase, we also follow it to simulate the distribution of the light probe that we aim to predict.

### 3.1.2 APPLYING NVDIFFREC

Starting with an initial tetra-grid, nvdiffrec learns the signed distance function (SDF) and the normal vector of each vertex using an MLP. It then adjusts the positions of vertices during training. Another MLP learns the texture, including the diffuse factor, specular factor, and roughness, which are used in calculating the BSDF reflection. At the end, the geometry is converted into the triangular mesh by Marching Tetrahedra, and the material is represented as 2D texture maps by XAtlas. For illumination, a cube map of the tensor is optimized during training. Since each input image is captured under varying illuminations in our assumption, we modify the pipeline to optimize the individual environment map for each different view.

After the decomposition, we obtain the triangular mesh, 2D texture, and environment map corresponding to each view. We further use this information to train an illumination MLP for predicting light probes in the second phase.

### 3.2 PHASE II: LEARNING ILLUMINATION MLP

In order to predict the illumination directly from the lighting of the scene, we train an illumination MLP to predict the color of a local light probe. Given the information about the target's geometry and material obtained in the first phase, we simulate the split sum approximation of specular reflection for the color and distribution of a light probe. For each sampled pixel from the input image, we multiply the color with the distribution to produce a light probe. Then, we stack these light probes to obtain the complete environment map, similar to the idea of multiplying the Gaussian spheres to approximate the integral of 'incident light' in most related work on BSDF reflection. To make the light probes close to physically-based rendering, we follow a split sum approximation used in nvdiffrec, simulating the light-probe representation as

$$\mathbf{P}_{\text{sample}} = c(\boldsymbol{c}, \boldsymbol{k}_d, \boldsymbol{k}_s, \boldsymbol{n}, \boldsymbol{r}, \omega_o) \cdot \mathbf{D}(\boldsymbol{n}, \boldsymbol{r}, \omega_o, \omega_i), \qquad (3)$$

where $\mathbf{P}_{\text{sample}}$ is an incident light probe produced from a sampled pixel and represented as a 3D tensor with the size of $(H, W, 3)$, while $c$ is the predicted color of the light probe from the illumination MLP given the MLP's input as the sampled pixel's color $\boldsymbol{c}$, diffusion factor $\boldsymbol{k}_d$, specular factor $\boldsymbol{k}_s$, normal vector $\boldsymbol{n}$, roughness $\boldsymbol{r}$ and viewing direction $\omega_o$ matched to the 3D geometry and texture. The viewing direction $\omega_o$ and incident light direction $\omega_i$ can be derived from the camera pose $[\boldsymbol{R}|t]$, the location of the reflected point on the surface of the target, and the location of incident light on the environment map. The tensor $\mathbf{D}$ is derived from the function $D(\omega_i, \omega_o)(\omega_i \cdot \boldsymbol{n})$ in Equation 2 and represented as a 2D tensor with the size of $(H, W)$. The process of the second phase is summarized in Figure 2.

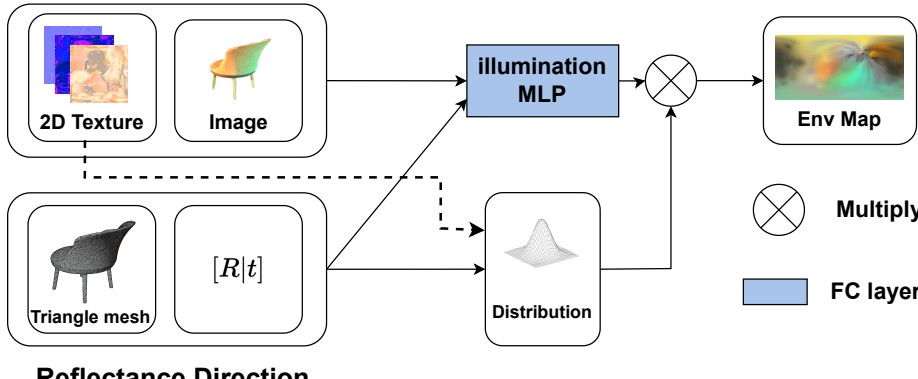

Figure 2: An overview of the second phase of our new physically-based approach.

### 3.2.1 ILLUMINATION MLP

We train an illumination MLP to predict the colors of incident light probes directly from the observed lighting of the scene. First, we sample pixels from the input image. By matching these pixels to the geometry and material learned from the first phase, we obtain the corresponding pixel's normal $\boldsymbol{n}$, diffusion $\boldsymbol{k}_d$, specular $\boldsymbol{k}_s$, and roughness $\boldsymbol{r}$. With the aforementioned information and the pixel color as the input, the illumination MLP is trained to predict the color of incident light probe as the output.

### 3.2.2 DISTRIBUTION OF LIGHT PROBE

We simulate the distribution of an incident light probe following the split sum approximation. The normal and roughness needed for the function **D** are also obtained by matching the pixels to the geometry and material estimated by nvdiffrec. While most recent methods related to the decomposition of reflection utilize Gaussian spheres to approximate the integral of the incident light, we represent the distribution of the light probe explicitly on the tensor equal to the size of the environment map

$$\mathbf{D}(\boldsymbol{n}, \boldsymbol{r}, \omega_o, \omega_i) = \frac{\boldsymbol{r}^2}{\pi \left(((\omega_o + \omega_i)/2)^2 \left(\boldsymbol{r}^2 - 1\right) + 1\right)^2} \left(\omega_o \cdot \boldsymbol{n}\right). \tag{4}$$

With this new approach, we can stack all the light probes in an instance and achieve better results.

### 3.2.3 STACKING LIGHT PROBES FROM SAMPLED DIRECTIONS

We use a weighted average to stack all the light probes produced by the sampled pixels. The weight we use while stacking the light probes is based on the light-probe distribution mentioned in the previous section:

$$\mathbf{P} = \sum_{n=1}^{N} \frac{c_n(\boldsymbol{c}, \boldsymbol{k}_d, \boldsymbol{k}_s, \boldsymbol{n}, \boldsymbol{r}, \omega_o) \cdot \mathbf{D}_n(\boldsymbol{n}, \boldsymbol{r}, \omega_o, \omega_i)}{\mathbf{D}_n(\boldsymbol{n}, \boldsymbol{r}, \omega_o, \omega_i)}, \tag{5}$$

where $N$ is the number of stacked light probes corresponding to $N$ sampled pixel. We set $N = 1024$ while training the illumination MLP in the second phase and set $N$ to $1/4$ the number of pixels on the test image inside the target mask during evaluation with the sample probability $P(\boldsymbol{r}) = |\boldsymbol{r} - 1| - 1$. We encourage sampling the light probes produced by pixels that have lower roughness since they offer more reliable information about the environment light in the corresponding directions.

### 3.3 LOSS FUNCTIONS

In the first phase, we adopt the same loss terms as in nvdiffrec, which includes mainly the L2 loss between the rendered image and the ground truth and other regularizers mentioned by (Munkberg et al., 2022).

In the second phase, the main loss is the image reconstruction loss between the stacked environment map and the optimized environment map obtained in the first phase. For better learning of the stacked environment map produced by sampled pixels, we propose a weighted L2 loss defined by

$$\text{Weighted L2 Loss} = \sum_{u,v}^{W,H} \frac{\mathbf{D}_{\text{sample}}^{(u,v)}}{\mathbf{D}_{\text{total}}^{(u,v)}} Y(\boldsymbol{c}_{u,v}) \left\| \boldsymbol{c}_{u,v} - \hat{\boldsymbol{c}}_{u,v} \right\|^2, \tag{6}$$

where $\mathbf{D}_{\text{sample}}$ is the summation of the sampled light probes' distributions as in Equation 5, which implies the importance of the area being sampled. The summation of the total light probes' distributions $\mathbf{D}_{\text{total}}$ is calculated as a weighted average by the number of light probes from each direction, which indicates the importance averaged from each direction of the light probe while stacking all of them produced by the pixels in the entire image. The main idea is to encourage the illumination MLP to learn high-contrast light colors like black and white with an intensity weight $Y(\boldsymbol{c}) = |0.299r + 0.587g + 0.114b - 0.5| + 0.5$.

## 4 EXPERIMENTS

### 4.1 DATASETS

We evaluate our proposed method on the NeRD (Boss et al., 2021a) datasets. We use the state-of-the-art method, Neural-PIL (Boss et al., 2021b), as the main baseline for comparison. The datasets we choose contain seven scenes categorized as follows: *i*) three synthetic scenes (Chair, Car, Globe) with each view relighted under varying illumination and camera pose, *ii*) two real-world scenes (Gnome, MotherChild) captured under varying illumination, and *iii*) two real-world scenes, one with fixed unknown illumination (Cape) and the other with relatively varying illumination (Head). Neural-PIL also tests their method on the NeRF dataset (Mildenhall et al., 2020), which is synthesized with fixed illumination. We do not test our method on the NeRF synthetic data because it only has fixed illumination, while our method aims to address a more challenging scenario of varying illuminations in multi-view images. As our method is based on nvdiffrec for the first phase, our results on the NeRF dataset would be identical to those shown in the nvdiffrec paper (Munkberg et al., 2022). We also compare our method with the other two neural rendering methods, NeRF and NeRD, as done by Neural-PIL. The qualitative results and the evaluation scores of NeRF and NeRD are taken from Neural-PIL's paper.

### 4.2 IMPLEMENTATION DETAILS

In the first phase, we follow most of the configurations in nvdiffrec. In order to obtain the fine-optimized environment map against each view, we set the batch size to one and training iterations to 40k rather than eight and 5k. Also, we adjust the learning rate of geometry to $0.015$, and material and light to $0.005$. All learning rates are descending exponentially from $1.0$ to $0.1$ of their values during the training. We set the resolution of the environment map to $(6, 128, 128, 3)$, which matches the resolution of our baseline. In the second phase, we train our illumination MLP with iterations depending on the size of the scenes. For NeRD synthetic scenes that contain 200 views, we set the iterations to 30k. For real-world scenes that contain around 100 views, we set the iterations to 30k. The learning rate of the illumination MLP is fixed at $0.0005$ during the training.

Both phases of training take within 3 hours on a single GTX 1080 Ti GPU, which is shorter than our baseline, Neural-PIL, that requires 22 hours on four RTX 2080 Ti GPUs, even with some pre-trained priors. Note that Neural-PIL's paper only presents the average scores and does not have the performance on individual scenes. Therefore, in experiments shown below, we also reproduce Neural-PIL, on 4 V100 GPUs with their released code and evaluate the individual result of each scene. Our code for the experiments will be released after the review process.

### 4.3 MAIN RESULTS

**Comparison with the State of the Art.** We show the experimental results of our method compared with the state-of-the-art baseline method and the other two prior methods. For the synthetic scenes under varying illuminations (Car, Chair, Globe), the results are shown in Table 1. We can

see that our method achieves competitive results on the Car and Chair scenes but fails on the reconstruction of the Globe scene because of the strong self-reflection that ends up being falsely copied into the material and thus causes the geometry defection. Table 2 shows the results of the scenes captured under varying illuminations in the real world (Gnome, MotherChild). Further, Table 3 presents the results of the scenes captured under relatively varying illuminations (Head) and fixed illumination (Cape) in the real world. For these real-world scenes, our method achieves a better score on PSNR but a slightly worse score on SSIM. Compared with the baseline method, which requires pre-trained with strong illumination before trying to match the most possible illumination, our method directly predicts the illumination from the lighting on the target and does not rely on any prior. Such an advantage allows our method to perform better on real-world conditions for higher PSNR. Our method reconstructs the target to the format of triangular mesh and 2D texture, which conforms to the most common rendering engine and hence is more efficient for training; however, the geometry reconstruction is also restricted to the complexity of triangular mesh and thus results in lower SSIM.

We also compare the inference time for lighting estimation from a single image in Table 4. Our method only requires a single-step inference that takes less than 1 second for a new view under unknown illumination, which is much faster than our baseline, requiring several minutes to optimize the illumination latent for 100 steps.

**Visualization.** We visualize some of our results in comparison with the baseline methods. The images of other methods, including Neraul-PIL, are taken from the original paper (Boss et al., 2021b). Figure 3 illustrates the decomposition of diffuse, specular, roughness, normal, illumination, and rendered images following the representations in their paper on the NeRD Car synthetic scene. Our method has some color baked into the environment map but is competitive with other methods overall. We also show the reconstruction of some novel-view synthesis images on the NeRD Cape real-world scene in Figure 4; our method exhibits better quality in visualization in contrast to the baseline method and others.

Table 1: Results on the NeRD synthetic varying-illumination scenes. The comparisons are evaluated using the average scores of NeRF (Mildenhall et al., 2020), NeRD (Boss et al., 2021a), and the main baseline method Neural-PIL (Boss et al., 2021b). The average results reported in the original paper of Neural-PIL are referred to as Neural-PIL (Ori.). We also reproduce the results using their released code, referred to as Neural-PIL (Rep.), for evaluating the scores of each individual scene.

| Method | Average scores | | Car | | Chair | | Globe | |
|---|---|---|---|---|---|---|---|---|
| | PSNR↑ | SSIM↑ | PSNR↑ | SSIM↑ | PSNR↑ | SSIM↑ | PSNR↑ | SSIM↑ |
| NeRF | 21.05 | 0.89 | - | - | - | - | - | - |
| NeRD | 27.96 | 0.95 | - | - | - | - | - | - |
| Neural-PIL (Ori.) | 29.24 | 0.96 | - | - | - | - | - | - |
| Neural-PIL (Rep.) | 26.33 | 0.88 | 27.74 | 0.92 | 29.51 | 0.92 | 21.95 | 0.81 |
| Ours | 27.51 | 0.94 | 29.09 | 0.96 | 29.61 | 0.95 | 23.83 | 0.90 |

Table 2: Results on the NeRD real-world varying-illuminations scenes.

| Method | Average scores | | Gnome | | MotherChild | |
|---|---|---|---|---|---|---|
| | PSNR↑ | SSIM↑ | PSNR↑ | SSIM↑ | PSNR↑ | SSIM↑ |
| NeR | 20.11 | 0.87 | - | - | - | - |
| NeRD | 25.81 | 0.95 | - | - | - | - |
| Neural-PIL (Ori.) | 26.23 | 0.95 | - | - | - | - |
| Neural-PIL (Rep.) | 22.92 | 0.87 | 23.73 | 0.81 | 22.11 | 0.92 |
| Ours | 27.48 | 0.94 | 25.25 | 0.90 | 29.67 | 0.97 |

**Ablation Study.** We conduct an ablation study on the NeRD Chiar scene to verify our proposed loss in Section 3.3. The weighted loss is designed to emphasize the importance of the sampled area and the intensity on the environment map; we compare the result produced by *i*) pure L2 loss, *ii*) L2 loss with intensity weight, *iii*) L2 loss with sample weight, and *iv*) L2 loss with sample weight and intensity weight in Table 5. L2 loss with sample weight gives better results than pure L2 loss

since it focuses on the area sampled among the environment map. L2 loss with intensity weight has little effect compared with pure L2 loss because the intensity weight through the entire environment map may not be related enough to the sampled area. However, combined with the sample weight, the intensity weight can accordingly encourage learning against extreme lighting. Therefore, this ablation study shows that our proposed L2 loss with sample and intensity weight is beneficial to achieve the best result.

Table 3: Results on the NeRD real-world fixed illumination scenes.

| Method | Average scores | | Head | | Cape | |
|---|---|---|---|---|---|---|
| | PSNR↑ | SSIM↑ | PSNR↑ | SSIM↑ | PSNR↑ | SSIM↑ |
| NeRF | 23.34 | 0.85 | - | - | - | - |
| NeRD | 23.86 | 0.88 | - | - | - | - |
| Neural-PIL (Ori.) | 23.95 | 0.90 | - | - | - | - |
| Neural-PIL (Rep.) | 22.18 | 0.78 | 23.14 | 0.85 | 21.21 | 0.71 |
| Ours | 24.00 | 0.87 | 24.01 | 0.89 | 23.98 | 0.85 |

Table 4: Comparison of inference time for lighting estimation from a single image.

| Method | Inference time (sec) | Inference steps |
|---|---|---|
| NeRD | $\approx$ an hour | 1000 |
| Neural-PIL | $\approx 300$ | 100 |
| ours | $< 1$ | 1 |

Table 5: Ablation results of different illumination losses on the NeRD Chair scene.

| Method | Evaluation scores | |
|---|---|---|
| | PSNR↑ | SSIM↑ |
| L2 loss | 28.68 | 0.95 |
| L2 loss + Intensity weight | 28.71 | 0.95 |
| L2 loss + Sample weight | 29.24 | 0.95 |
| L2 loss + Sample weight + Intensity weight | 29.61 | 0.95 |

## 5 CONCLUSION

We have proposed a method for learning varying environment maps from multi-view images of the target captured under different illuminations. We first apply the fixed-illumination decomposition method, nvdiffrec, to our varying-illumination scenario for the geometry and material of the target. Then, we train an illumination MLP to predict the color of the light probes sampled from the target. With the distribution that follows physically-based rendering for each light probe, we produce a complete environment from the target image depending on its illumination. Unlike the existing methods that rely on strong illumination prior, our method learns the relation between the lighting on the target and the environment map. Furthermore, our method is more efficient for both training and inference on learning to predict varying environment maps from single images.

**Limitations.** The accuracy of the decomposed material is always an issue for reflection decomposition under both fixed and varying illuminations. A common issue would be the lighting baked into the material or the material baked into illumination. Our method applies nvdiffrec to solve the decomposition under fixed illumination for the geometry and material of the target, which could be improved on the accuracy of estimating the material. Without assuming strong illumination prior or using iterative optimization to estimate the environment map, our method predicts the illumination from lighting on the target with a representation simulating physically-based rendering. While the physically-based simulation characterizes a more realistic relation between lighting on the target and the illumination, it might also miss details in producing the complete environment map from sampled light probes, which is worth further exploration.

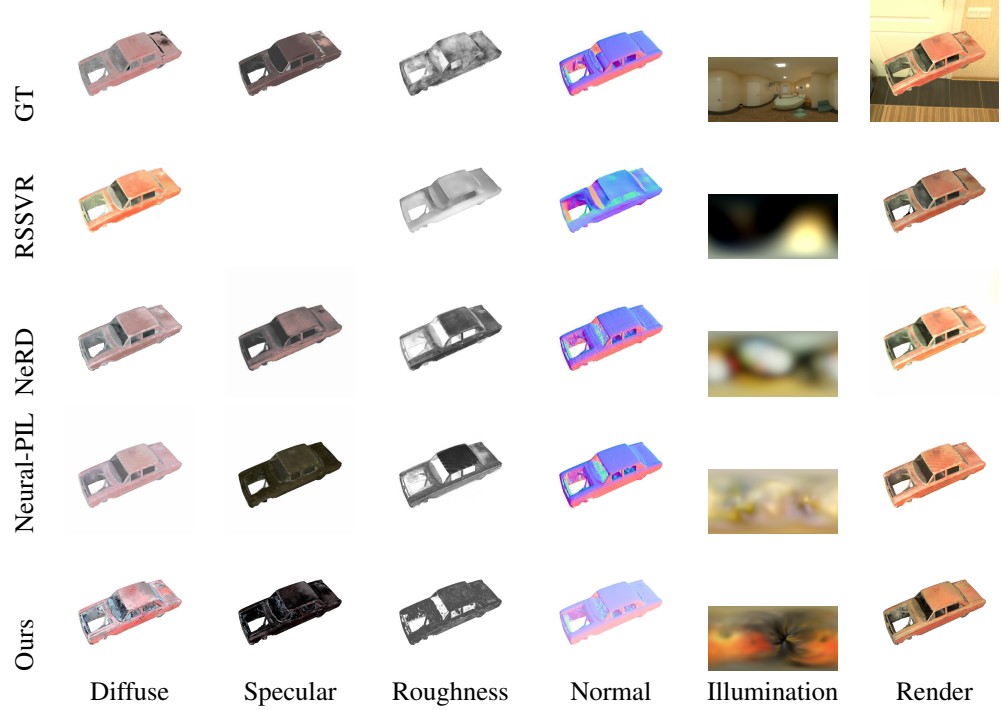

Figure 3: Visualization of decomposition. The qualitative comparisons of Neural-PIL, RSSVR Li et al. (2018), NeRF and NeRD.

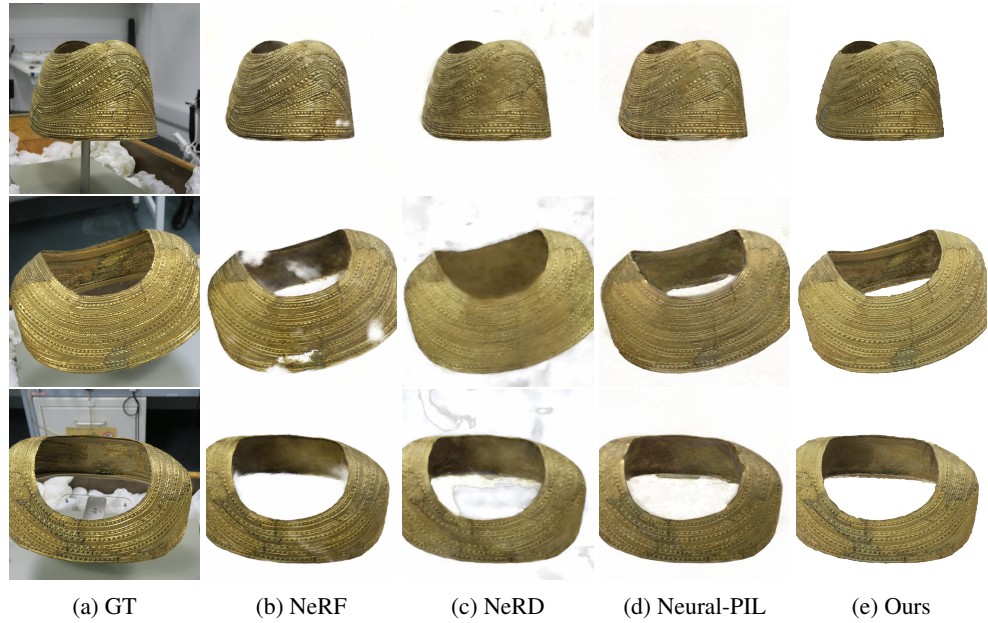

Figure 4: Visualization of Gold Cape.

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
