# [Supplementary Material of Paper 984] Knowledge Distillation for Predicting Varying Environment Maps from Single Images

## More Qualitative Results

We present more qualitative results of our method performed on the test images of the NeRD scenes. The first row of each figure shows the ground-truth lighting results in different views, and the second rows shows the results rendered by our method.

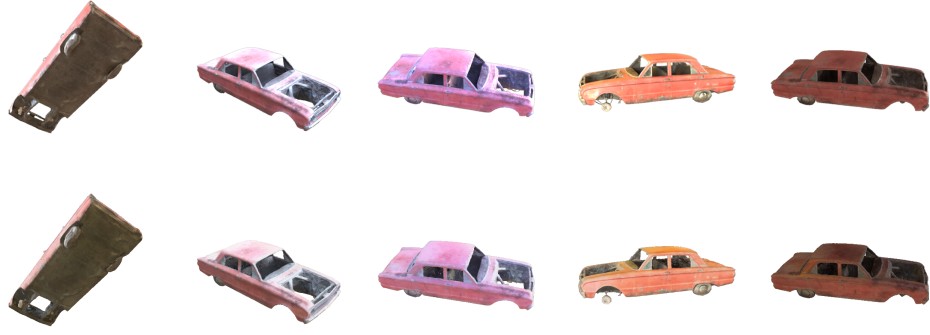

Figure 1: Visualization of Car.

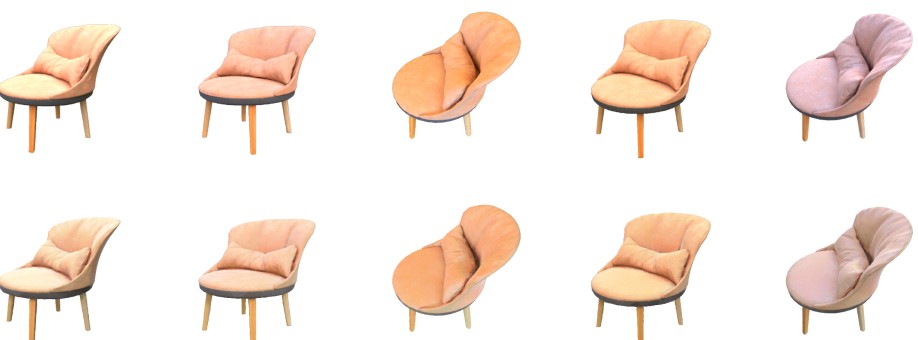

Figure 2: Visualization of Chair.

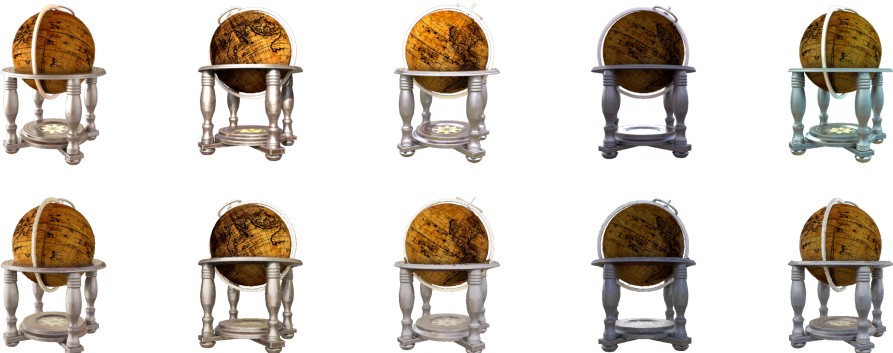

Figure 3: Visualization of Globe.

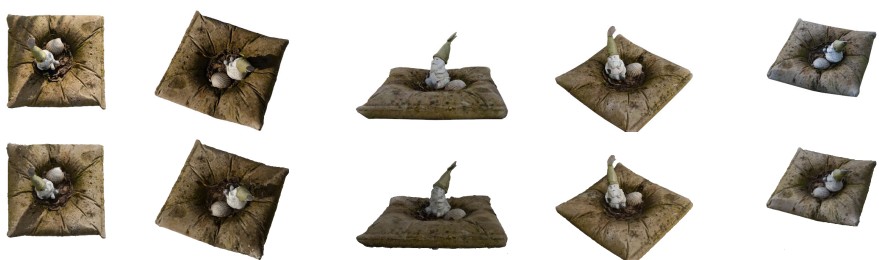

Figure 4: Visualization of Gnome.

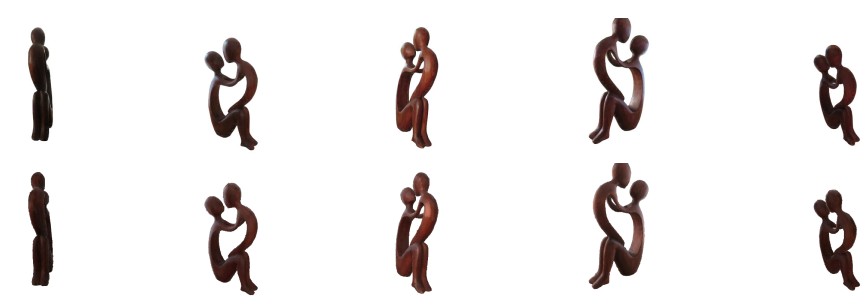

Figure 5: Visualization of MotherChild.

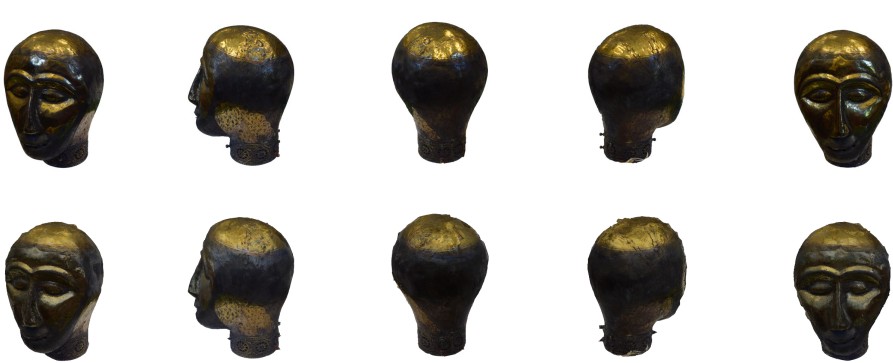

Figure 6: Visualization of Head.

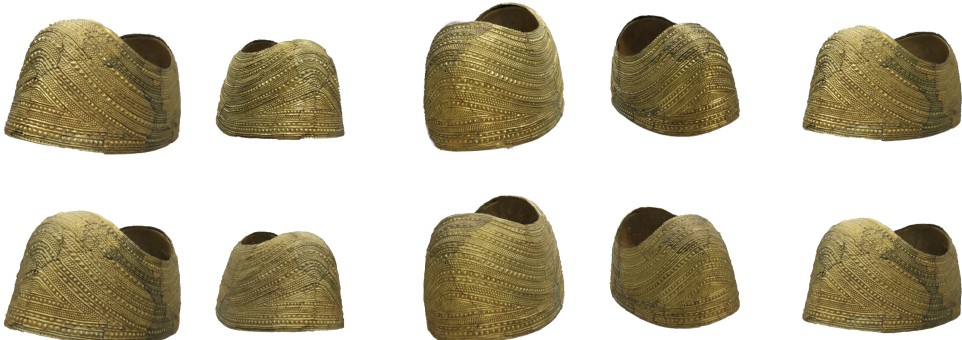

Figure 7: Visualization of Cape.