# OpenReview forum: "Knowledge Distillation for Predicting Varying Environment Maps from Single Images"
_ICLR.cc/2024/Conference — ICLR 2024 Conference Withdrawn Submission_

### Official Review · Reviewer_Pvo6 · 2023-10-30

**Soundness:** 2 fair
**Presentation:** 3 good
**Contribution:** 2 fair
**Rating:** 5
**Confidence:** 4

**Summary:**

This paper proposes an novel approach for predicting environment map under complicated circumstances by introducing knowledge distillation. Using the backbone of Neural-PIL, the proposed algorithm utilizes information acquired from Neural-PIL to form light distribution and stacked light probes under the guidance of the illumination MLP in order to calibrate the loss function. Experiments are conducted on NeRD dataset compared with several baselines, as well as checking the method with additional ablation studies.

**Strengths:**

1. The proposed method is simple, while the structure of which shows some interesting motivation and insights.
2. The related work show section is concrete, which shows the necessary backgrounds of the field.

**Weaknesses:**

1. This paper’s illustration of its proposed method is not sufficient. The author uses a simple verbal explanation of the methodology of the proposed method with overview graphs, without specific instructions on the detailed specification.

2. One of my biggest concerns about this paper is its experiment, which is not convincing enough. All tests are only conducted on a single dataset, with some experimental results being either unsatisfying or missing.

3. The section 3.2.3, which introduces stacking light probes, is vague and not well organized. For example, as illustrated in eq. 5, ‘P’ should be a vector with the size of 3, while it is interpreted as a (u,v) matrix in section 3.3.

4. The experiment seems confusing to me. As shown in the paper, the results of the proposed approach are not good enough on the average scores compared with other baselines. For instance, the average scores of the proposed method on both metrics are not the best, while the score on PSNR is worse than that of NeRD and Neural-PIL(Ori.).

5. When it comes to the statistics of specific classes, the results of NeRF and NeRD are not applicable. Considering that this dataset was originally been used by NeRD, it would be more suitable to test these baselines on every class in your tables and reproduce them if you lack these results in the Nerual-PIL’s paper. If the reproduction is too time-consuming, you should find other baselines, as only one baseline is not enough to show your work’s effectiveness.

6. The reproduced results on Neural-PIL are much lower than the original ones. Please double-check your code’s correctness to minimize this gap to an acceptable level.

7. The title of NeRF misses an “F” in table. 2.

**Questions:**

What are the meanings of ‘c’ and ‘c^(hat)’ in eq. 6? Please make sure that every symbol in your equations is well defined.

---

### Official Review · Reviewer_nhSg · 2023-10-30

**Soundness:** 3 good
**Presentation:** 2 fair
**Contribution:** 2 fair
**Rating:** 5
**Confidence:** 4

**Summary:**

This work improves the nvdiffrec method to decompose the object geometry, material and scene illumination given multi-view images with varying illuminations. The key strength is to optimize view-dependent environment maps during scene decomposition, and an illumination mlp to render environment maps under single input image with different illuminations.

**Strengths:**

Extending nvdiffrec to a more practical scenarios with varied input illumination.
A compact pipeline to achieve decomposition and rendering with varied illuminations.

**Weaknesses:**

Handling varied appearances has been widely investigated in NeRF methods, such as [1]. The problem from NeRF to [1] is very similar to the proposed method. Instead of comparing to NeRF, a very straight forward approach is to apply [1] (with only appearance embeddings), and render novel views for the task of novel view synthesis. And if we want to measure the accuracy of decompositions, we can also apply [1] to obtain for each training view the rendering under varied illuminations, and then apply nvdiffrec on the rendered training views (with adjusted illuminations) to obtain the final decomposition. I think missing this naive and straight forward baseline in the comparison makes the advantage of the proposed method unclear.

I would increase my score if the above problem can be properly addressed.

[1] NeRF in the Wild.

**Questions:**

N.A.

---

### Official Review · Reviewer_cpaY · 2023-11-01

**Soundness:** 3 good
**Presentation:** 3 good
**Contribution:** 3 good
**Rating:** 6
**Confidence:** 3

**Summary:**

The authors introduce a novel method for the reconstruction and reflection decomposition in multi-view image collections under varying illumination. Starting with a construction process to determine geometry and material, they subsequently train an illumination MLP to predict the environment map from a target image. Their approach is based on nvdiffrec, in its first stage produces a triangular mesh, neural texture, and individual environment maps. This provides the groundwork for the second-stage training of the illumination MLP. The authors report improvements in performance over both synthetic and real datasets. Moreover, they accomplish this in a more time-efficient manner than other existing techniques.

**Strengths:**

1. This paper extends nvdiffrec, which was initially intended for fixed lighting conditions. Their adaptation to varying lighting conditions presents a good step forward in the domain.
2. Unlike many contemporary methods that deploy Gaussian spheres to approximate the integral of the incident light for reflection decomposition, this work explicitly represents the distribution of the light probe on a tensor that matches the environment map's size. This could potentially offer a more accurate representation of real-world scenarios.
3. The method presented is more efficient in both the training and inference stages than alternative methods requiring iterative optimization. This efficiency is crucial for practical applications in real-world scenarios.

**Weaknesses:**

1. It's puzzling why the authors did not illustrate a direct comparison with other baselines (e.g. traditional, NeRF based, spherical gaussian based methods). Such a comparison could have provided a more comprehensive understanding of their method's advantages or shortcomings.
2. The use of a physically-based simulation might occasionally omit details when generating the complete environment map from sampled light probes. This could be a potential direction for further research.
3. The paper doesn't sufficiently address the generalization capability of their method across diverse scenarios. There is an underlying concern: is the proposed method mainly efficacious in scenarios where nvdiffrec performs well?

**Questions:**

1.Are there any insights or preliminary results on how the method generalizes across different types of scenes or lighting conditions not covered in the datasets used?
2. Can the authors comment on scenarios where nvdiffrec might not perform optimally and how this might affect the robustness of their proposed method?

---

### Official Review · Reviewer_zAh2 · 2023-11-05

**Soundness:** 2 fair
**Presentation:** 2 fair
**Contribution:** 2 fair
**Rating:** 3
**Confidence:** 3

**Summary:**

The paper proposed a two-stage approach to learn a direct mapping from an image to its environment map.
The first stage involves learning the scene's geometry and texture as well as the environment map using a differentiable rendering framework (by using the established nvdiffrec framework).
The second stage trains an MLP to directly map a single image to its environment map by learning (distilling) from the model trained in the first stage.
The experiments show that the method is comparable to the state of the art in terms of PSNR and SSIM metrics on a few synthetic and real-world datasets while being much more efficient.

**Strengths:**

The idea of learning to directly predict the environment map given an image by distilling what has been learned in the first stage seems valid.

**Weaknesses:**

1. The writing of the paper could be improved. The first stage of the proposed method built on nvdiffrec, but it may still be useful to explain how the first stage works with more details. Section 3.1.1 lists the rendering equations, and section 3.1.2 explains how nvdiffrec is applied. But both sections missed the opportunity to explain clearly how the first stage works. For example, what's the loss function? How are the rendering equations Eq(1) and Eq (2) involved in the loss function? What are being optimized? Section 3.1.2 somewhat tries to explain what has been optimized in texts, but how does the reader relate the texts with the equations? Both Figure 1 and Figure 2 can use a more detailed caption.

2. It's very hard to follow the description of stage 2. (1) Maybe a little motivation about Eq. 4 could help? (2) Section 3.3, I found it really hard to follow: "where D_sample is the summation of the sampled light probes’ distributions as in Equation 5 ...  D_total is calculated as a weighted average by the number of light probes from each direction". Maybe show the equation of D_sample and D_total can help. The consistency of the notation could also be improved. In Eq. (5), subscript is used to indicate the sample index, and in Eq. (6), superscript (u, v) is used to indicate the sampling location.

3. Experiments are not very thorough. The proposed method is mostly only compared against the reproduced Neural-PIL which has inferior performance compared to the original Neural-PIL as can be seen in Table 1 (original implementation registers average PSNR of 29.24 and SSIM 0.96 whereas the reproduced version has PSNR of 26.33 and SSIM 0.88). That said, it's not clear if the author's method is actually better than the original version of Neural-PIL. In Table 1, not sure why NeRF, NeRD, and Neural-PIL (Orig.) do not have results on Car Chair, and Globe datasets. In Table 2, "NeR" (typo? should be NeRF), NeRD, and Neural-PIL (Orig.,) do not have results on Gnome and MotherChild dataset. In Table 3, NeRF, NeRD, and Neural-PIL (Ori.) do not have results on Head, and Cape dataset.

**Questions:**

See Weakness section above, it can greatly help if the authors could clarify the method more -- especially the second phase which seems to be the main differentiator of the work.